# Cognitive Difficulties, Psychological Symptoms, and Long Lasting Somatic Complaints in Adolescents with Previous SARS-CoV-2 Infection: A Telehealth Cross-Sectional Pilot Study

**DOI:** 10.3390/brainsci12080969

**Published:** 2022-07-23

**Authors:** Samuela Tarantino, Sonia Graziano, Chiara Carducci, Rosaria Giampaolo, Teresa Grimaldi Capitello

**Affiliations:** 1Department of Neurological Sciences, Unit of Clinical Psychology, Bambino Gesù Children’s Hospital, IRCCS, 00165 Rome, Italy; sonia.graziano@opbg.net (S.G.); chiara.carducci@opbg.net (C.C.); teresa.grimaldi@opbg.net (T.G.C.); 2Department of Paediatric Medicine, Bambino Gesù Children’s Hospital, IRCCS, 00165 Rome, Italy; rosaria.giampaolo@opbg.net

**Keywords:** COVID-19, adolescents, anxiety, depression, cognitive functioning, memory, attention skills

## Abstract

Background. Few studies have evaluated cognitive functioning and mental health in children and adolescents who contracted the SARS-CoV-2 infection. We investigated the prevalence and association of neuropsychological difficulties, psychological symptoms, and self-reported long-COVID complaints in a sample of adolescents. Methods. Thirty-one adolescents infected by COVID-19 within 3–6 months prior to the assessment were included. Neuropsychological difficulties, psychological symptoms, and self-reported long-COVID complaints were evaluated using a checklist and a battery of multiple standardized measures, using a telehealth procedure. Symptoms during the infection were also detected. Results. We included 31 adolescents (23 girls, 8 boys; mean age 14.1, SD = 2). We found borderline scores in 32.3% and 45.2% of our sample for phonemic and category fluency, respectively. A high percentage of participants showed symptoms of depression (80.6%) and anxiety (61.3%). Fifty-eight percent reported at least one long-COVID symptom. The most common symptoms were headache and attention problems (58%). Subjects presenting numbness/weakness, fatigue, brain fog, or attention problems had higher scores in depression, anxiety, and post-traumatic stress symptoms (*p* ≤ 0.05). Conclusion. This is a pilot study limited by the lack of control group. However, we found that cognitive, psychological, and physical symptoms were very common among adolescents recovered from COVID-19.

## 1. Introduction

SARS-CoV-2 is now recognized as a multi-organ disease [1,2]. The clinical features and severity of the disease may widely vary among patients, with some experiencing severe symptoms and need of hospitalization, and others developing mild or even asymptomatic infections [3,4,5]. Increasing attention has been paid to the long-term effects of COVID-19 infection. The post-acute sequelae of COVID-19, termed “long-COVID” is used to describe persistent multi-organ symptoms that continue or develop after acute COVID-19 infection and are not explained by an alternative diagnosis [6,7,8]. Long-lasting symptoms are frequent, even in individuals with mild symptoms [9]. To date, there is no clear agreement on the definition or duration for this syndrome. According to NICE guidelines, long-COVID includes both on-going symptomatic COVID-19 (from 4 to 12 weeks after the infection) and post-COVID-19 syndrome (12 weeks or more after acute COVID-19) [10,11]. Long-COVID is characterized not only by physical symptoms (e.g., fatigue, lack of stamina, cough, difficulty breathing, muscle pain, numbness/weakness, smell and taste disorders), but also by cognitive and psychological problems [8,12,13,14,15]. The symptoms, such as attention and memory disorders, brain fog, fatigue, anxiety, and depression, may be common components of long-COVID syndrome, with symptoms even more common in the long term (six or more months post infection) than the mid term (three to six months) [14]. This led some authors to suggest that these symptoms are more likely to develop than to persist after the infection [9,14]. Neuropsychological studies on adults found persistent objective cognitive problems with executive functions, memory, and attention problems for several months after recovery from COVID-19 [16,17,18].

Increasing research has explored the prevalence and impact of long-COVID on health and quality of life in pediatric age [10,11,19,20,21]. The most commonly reported symptoms resemble those described in the adult populations, with a prevalence of headache ranging from 3 to 80%, fatigue ranging from 3 to 87%, and concentration difficulties ranging from 2 to 81% [22,23].

Regarding the children and adolescents who contracted the viral infection, the data concerning the cognitive functioning and mental health are limited [24,25,26,27,28]. To the best of our knowledge, only two studies have evaluated cognitive problems in children and adolescents recovered from SARS-CoV-2 infection, and they did not explore the association with physical long-COVID complaints or psychological symptoms or included patients with a large age range [27,28]. No studies analyzed the possible association between cognitive and psychological problems with self-reported long-COVID complaints among adolescents.

Our study aimed to investigate the prevalence of neuropsychological difficulties, psychological symptoms, and self-reported long-COVID complaints in a sample of adolescents (3–6 months after recovery from infection), using a screening procedure via telehealth. We studied the possible differences in cognitive and psychological functioning: (1) between adolescents who had symptoms during the infection and those who were asymptomatic; and (2) between adolescents who reported long-COVID symptoms and those who did not.

We hypothesized that: (1) adolescents recovered from SARS-CoV-2 infection would report a high prevalence of neuropsychological difficulties, psychological symptoms and long-COVID complaints; (2) cognitive difficulties, psychological symptoms, and self-reported long-COVID complaints would be higher in the subjects reporting symptoms during the infection; and (3) neuropsychological performance would be lower in the adolescents who described symptoms during the infection and in those who reported at least one long-COVID symptom; on the other hand, we hypothesized higher psychological symptoms in these adolescents’ subgroups.

## 2. Methods

We conducted a cross-sectional pilot study documenting the neuropsychological and psychological effects of the SARS-CoV-2 infection in adolescents, using telehealth. The data were collected between July 2021 and December 2021. The participants were recruited among the patients referred for a consultation to Bambino Gesù Children’s Hospital in Rome.

The sample included adolescents aged 12–18 years infected by COVID-19 within 3 to 6 months prior to the assessment. We excluded: (1) adolescents with the presence of any medical conditions or other neurological diseases that may interfere with normal cognitive development; (2) those with a diagnosis of intellectual disabilities, specific learning disorders, or other psychiatric disorders; (3) patients who had reported attention and memory difficulties prior to the infection. All of the subjects had to be able to read, understand, and answer every item of the screening. None had received any vaccination for COVID-19.

The participants completed the measures electronically. A single 30–40-min video session was carried out, using the Zoom online video platforms for each of the participants. All of the measures were completed using the screen-sharing function on Zoom. The participants were asked to use a computer and to find a quiet, private place, using headphones and microphone. In addition, having a stable internet connection and familiarity using the internet was requested. Two psychologists experienced in child psychology administered the psychological and neuropsychological screening measures.

We considered the following variables: (a) cognitive difficulties: memory, executive functions, attention, and processing speed; (b) psychological symptoms: stress and post-traumatic symptoms, depression, anxiety; (c) long-COVID symptoms; (d) feasibility and satisfaction with the telehealth screening procedure.

Written informed consent was obtained from the parents and participants. This study was approved by the Bambino Gesù Ethical Committee.

### 2.1. Measures

#### 2.1.1. Cognitive Measures

-Memory. The “Forward and Backward Digit Span” subtest of the Italian BVN 12–18 (Neuropsychological evaluation battery for adolescents) was used to explore the verbal short-term and working memory [29]. It evaluates the span of numbers correctly remembered by the subject in backward and forward order. The “Word List Recall Task”, which consists of a 12- word list, evaluates learning and long-term verbal memory. The participant is required to recall as many words as possible for a maximum of eight trials. Delayed verbal memory is assessed by asking the subject to remember the same list of words after 30 min.-Executive functions. The “Verbal Fluency” subtest of BVN 12–18 evaluates executive functions. It consists of category fluency and phonemic fluency to test the speed of access to the lexicon, self-monitoring, and selective inhibition [29]. The participants were given one minute to produce as many words as possible within a specific category (i.e., animal, fruits) or starting with a given letter. The scores were computed for each category (or given letter) and a total as a summary by adding all of the scores together.-Attention and processing speed. The “Symbol Digit Modalities Test” (SDMT) [30,31,32], provides a quick screening for organic cerebral dysfunction in both children and adults. Using a reference key, the subject has 90 s to pair specific numbers with given geometric figures. Responses can be written or given orally. In our study, we administered the oral version of the test.

The raw scores of BVN 12–18 and DSMT were converted into standard scores [29,30,31,32].

#### 2.1.2. Psychological Measures

-Perceived rating stress. A stress rating was administered, asking the participants to evaluate their perceived level of stress on an ad hoc, 10-point Likert scale, ranging from 1 (not stressed at all) to 10 (extremely stressed).-Post-Traumatic Stress Symptoms. The “Trauma symptoms checklist for children” (TSCC) is a self-administered measure of post-traumatic stress and related psychological symptomatology (anxiety, depression, anger, dissociation) in children aged 8–16 years who have experienced traumatic events, such as physical or sexual abuse, major loss, or natural disasters [33]. We administered the eight items of the Post-Traumatic Stress Scale (PTS). The raw scores were converted into T-scores [33].-Depression and anxiety. The Patient Health Questionnaire-9 (PHQ-9) and the Generalized Anxiety Disorder-7 (GAD-7) are brief, self-administered measures of depressive symptoms and anxiety [34,35,36]. Each symptom/item is rated, according to frequency, using a 4-point rating scale over the past two weeks. The PHQ-9 includes a question about suicidal ideation (#9) which asks about “thoughts that you would be better off dead, or of hurting yourself,” over the last 14 days. The total scores range, respectively, from 0 to 27 in the PHQ-9 and 0 to 21 in the GAD-7, with higher scores indicating more severe depressive and anxiety symptoms. The severity of the depressive symptoms is categorized as “no symptoms” (0–4), “mild symptoms” (5–9), “moderate symptoms” (10–14), and “severe symptoms” (≥15) [37].

#### 2.1.3. Symptoms during SARS-CoV-2 Infection and Self-Reported Long-COVID Symptoms

The presence of symptoms during acute SARS-CoV-2 infection was explored by a checklist to assess the following features: presence of fever; cough; difficulty breathing; need for oxygen; and hospitalization. The presence of long-COVID subjective complaints was evaluated by an ad hoc checklist that included a range of somatic and cognitive problems, developed based on common symptoms described in previous studies in adults [8,13,14,15,38]: (a) headache; (b) abdominal pain; (c) numbness/weakness; (d) fatigue; (e) smell and taste disorders; (f) attention problems; (g) memory problems; (h) brain fog.

#### 2.1.4. Feasibility and Satisfaction

The feasibility of participation and satisfaction with the intervention were assessed using an ad hoc 4-point Likert (1 to 4) scale at the end of the video session. Two feasibility questions were asked (How practical/easy was it to participate in the screening? Do you think the screening should continue?) and two items rated satisfaction (Did you find it helpful to participate in this screening? How satisfied were you with this screening?).

#### 2.1.5. Statistical Methods

The statistical analysis was performed using SPSS 22.0 software (Statistical Package for the Social Sciences).

According to the aim of our study, the participants were classified into: a) symptomatic (subjects describing one or more symptoms during the infection) and asymptomatic (subjects who did not have symptoms during the infection) subgroups; b) long-COVID (presence of at least one self-reported long-COVID symptoms) and no long-COVID (absence of long-COVID symptoms) subgroups.

We used descriptive statistics, expressed as means, SD and percentages, to describe the basic features of our sample.

A *T*-test was used to compare the psychological and neuropsychological scores between the symptomatic and asymptomatic subjects and between the adolescents who described long-COVID symptoms and those who did not.

The level of statistical significance was set at *p* < 0.05.

## 3. Results

Forty-one adolescents were examined for eligibility. Among them, five were excluded because of a diagnosis of learning disabilities. Six subjects declared that they were not interested in the study (*n* = 2) or did not have time (*n* = 4). The final sample included 31 adolescents (23 girls, 8 boys; mean age 14.1, SD = 2). No subjects had missing data.

All of the participants were students (52% in high-school and 48% in secondary school). None had undergone a previous psychological/neuropsychological evaluation.

Ten adolescents (32%) did not report any symptoms during the acute infection, while 21 (68%) had been affected by mild symptoms (e.g., fever, cough, respiratory difficulties). The most common symptom reported during the acute infection was fever. Only one participant was hospitalized and required oxygen therapy for a week. After the acute infection, all of the adolescents could continue normal daily activities.

Our data showed normal performance in both the short- and long-term memory (BVN-Span of numbers and words list recall) and processing speed (SDMT) assessments in the majority of the participants (64%). Borderline scores were found in 32.3% and 45.2% of our sample for phonemic and category fluency, respectively (BVN subtests). About 20% of the adolescents scored in the clinical range in these tasks (19.3% and 9.7, respectively). The neuropsychological findings are described in Table 1-Panel A.

The average stress rating was 4.8 (SD = 2.7) out of 10, indicating mild levels of distress. No significant post-traumatic symptoms were found (TSCC-A/PTS mean score = 6.1; SD = 3.8) and only one adolescent showed clinical symptoms. A high percentage of the participants showed symptoms of depression and anxiety, with 80.6% and 61.3% endorsing scores above the clinical cut-off, respectively. In particular, 32% scored in the moderate to severe range (see Table 1-Panel B).

In terms of suicidal/self-harm ideation, 16.2% of adolescents endorsed item 9 of the PHQ-9 measure for depression screening. For those who endorsed this item, a clinical in-person assessment was scheduled.

Among our sample, 58% reported at least one long-COVID symptom (Figure 1). These symptoms showed a high prevalence in both the symptomatic and asymptomatic subgroups (>80%).

The most common symptoms listed in the checklist were headache and attention problems (58%); on the other hand, the least common symptoms were smell and taste disorders.

When we explored the differences in cognitive and psychological symptoms between the symptomatic and asymptomatic participants, we found higher anxiety (*p* = 0.04) and post-traumatic symptoms (*p* = 0.02) in the adolescents who reported “difficulty in breathing”; on the other hand, they showed a lower performance in the SDMT and “category” fluency tasks (*p* = 0.01). No other differences were found in the scores on mental health scales and cognitive profile between the two groups (*p* ≥ 0.05).

Moreover, the *T*-test analysis showed statistical differences in the cognitive performance and the scores on the mental health scales between the long-COVID and no long-COVID subgroups. The subjects presenting numbness/weakness, fatigue, brain fog, or attention problems had higher scores in the depression, anxiety, and post-traumatic stress symptoms (*p* ≤ 0.05). Moreover, the adolescents who complained of headache had higher scores in anxiety (*p* = 0.02) and depression (*p* = 0.03).

Regarding the neuropsychological data, we found lower mean scores in the verbal short-term memory (BVN-Forward span of numbers) in subjects who described fatigue, brain fog, and reported memory problems (*p* ≤ 0.05). Our data also showed lower scores in the BVN-category fluency in the subjects reporting memory problems (*p* = 0.04).

The average feasibility across the participants was 3.3, average satisfaction was 3.1. Overall, these results suggested that the telehealth screening was both feasible and helpful.

## 4. Discussion

This cross-sectional pilot study investigated the prevalence of neuropsychological difficulties, psychological symptoms, and self-reported long-COVID complaints, in adolescents affected by the SARS-CoV-2 infection, using a telehealth screening procedure. Overall, this telehealth screening procedure was both feasible and considered helpful by the subjects. This confirmed the current telehealth literature providing consistent evidence of its benefits for communication and counseling [39,40]. Increasing studies have demonstrated telehealth’s feasibility and a high level of satisfaction from providers, patients, and families [41].

In our sample, the cognitive, psychological, and long-COVID somatic complaints were very common among adolescents recovered from COVID-19. There is an increasing interest in studying the sequelae of SARS-CoV-2 infection in children and adolescents; the results, however, are still conflicting and far from conclusive. The prevalence of pediatric long-COVID has shown a large variability between studies, ranging from 4 to 66% depending on the symptoms explored, sample size, median age of the included populations, evaluation modalities (a face-to-face or online assessment) and the duration of follow-up [22,23].

Our data confirmed the previous literature, showing that long-COVID complaints may be very common in pediatric age [6,7,8]. We found that more than half of the adolescents who recovered from SARS-CoV-2 infection reported mild symptoms of long-COVID, 3 to 6 months after the infection. These findings are consistent with the recent literature, showing the persistence of at least one complaint, even after 120 days from the initial diagnosis, regardless of the clinical features of the illness. According to previous findings, the most common somatic complaints described in our sample were headache and numbness/weakness, while subjective attention difficulty was the most reported cognitive symptom [20,21,22].

During the initial phase of the pandemic, the attention of researchers focused on the psychological adaptation of children and adolescents to the pandemic. Over time, more attention has been paid to the psychological consequences of the SARS-CoV-2 infection. In accordance with previous data, which demonstrated an increased risk of depression in the patients recovered from COVID-19, our data show a very high of rate of anxiety and, especially, depression [24,25,26]. Of note, among our adolescents, 16.2% of the sample scored positive for suicidal or self-harm ideation.

Little is known about the prevalence, the pattern, and the association with illness features of the cognitive impairments after SARS-CoV-2 infection. Only two studies have explored the cognitive abilities of children and adolescents recovered from SARS-CoV-2, and they display a large variability in the methodological design, sample involved, and tools used. Frolli et al. found a low cognitive and executive function performance, especially in the symptomatic adolescents, mainly those who required hospitalization [28]. In a small case-series cohort study on subjects (*n* = 9) ranging from 4 to 21 years, Morrow et al. demonstrated normal working memory, executive functions, verbal ability, and visual reasoning, while difficulties in sustained auditory and attention tasks and anxiety and/or mood concerns emerged (which were pre-existing to the virus in most of the patients) [27].

Although our study did not reveal any cognitive impairment, except for a weakness in executive functions, low neuropsychological performance was found in the adolescents who reported symptoms during the acute infection, and in those who reported subjective long-COVID complaints. In accordance with previous findings that suggest a potential link between restricted oxygen delivery to the brain and cognitive impairments [42,43,44], we found low-processing speed and executive functions in the adolescents with respiratory difficulties when they were ill. In this group, we also found high rates of anxiety and post-traumatic stress symptoms, which are also common in acute respiratory distress syndrome patients [42,45].

This is the first study to compare the cognitive performance and psychological symptoms between adolescents with and without long-COVID complaints. In our sample, the self-reported symptoms, such as fatigue, brain fog, or self-reported memory problems, were associated with both anxiety and depression, but also with low performance on the memory or executive function tasks.

The role of systemic inflammation in both cognitive impairments and psychiatric problems in the patients recovered from COVID-19 has been previously reported. In a study of Zhu et al., a possible association between the underlying inflammatory processes (measured with C-Reactive Protein-CRP) and sustained attention was described [46]. With respect to mental health, Yuan et al. described significantly higher depression in the convalescent COVID-19 patients with a higher Neutrophil-to-Lymphocyte Ratio (NLR) [47].

The exact underlying mechanisms causing the comorbidity between physical long-COVID complaints, psychiatric symptoms, and cognitive difficulties remain unclear. In a study on adults, the immune response and systemic inflammation (measured by systemic immune-inflammation index-SII) predicted cognitive impairment and severity of depressive symptoms at the three-months follow-up, thus confirming the possible connection between depression, inflammation, and cognition [48]. In particular, the prevalence of cognitive impairment was influenced both by the presence of systemic inflammation and by psychopathology [48]. Several authors, on the other hand, have hypothesized that the psychiatric and cognitive consequences of the SARS-CoV-2 infection may be due to factors not related to the immune response to the virus itself, but may be exacerbated by the stress related to the pandemic. Similar levels of unhappiness (98.7%), listlessness (80.7%), and tenseness (86.4%) in adolescents who were both seropositive and seronegative for COVID-19 were described among adolescents [49].

There is also evidence that, while the somatic symptoms due to SARS-CoV-2 tend to regress in a few months, the mental health problems may persist for a longer time [23]. According to these findings, it is possible that, in our sample, the anxiety and depression may be related to the pandemic and may lead to a vicious cycle, where anxiety and depression may negatively impact on cognitive performance and somatic complaints (e.g., brain fog and fatigue), which in turn may heighten anxiety and depression.

Moreover, somatic complaints, such as headache, together with anxiety and mood disorders are also very common in non-COVID-19 samples. A nationwide Italian study, conducted shortly before the outbreak of the pandemic, reported recurrent headache in almost 45% among the population (15-year-old adolescents) [50]. As for mental health, over the last few decades, several studies described an increasing risk of depression and anxiety among children and young people [51,52]. As reported in 2017 by WHO, depressive disorders and anxiety are, respectively, the third and the fifth most frequent cause of adolescent disability-adjusted life-years lost. The literature data on non-COVID-19 samples described a relatively high rate of suicidal ideation. In a study of Souza et al., suicidal ideation in early adolescence showed a rate of 14.1%, which is very close to the rate found in our study [53].

The studies comparing the COVID-19 samples to non-COVID-19 ones are required, in order to better understand the impact of COVID-19 on cognitive performances and psychological symptoms.

This study had several limitations. It included a small sample size and there was not a matched control group, therefore, the study may not be representative of the whole population of adolescents recovered from SARS-CoV-2 infection. In particular, the absence of a control group may represent a selection bias in order to be able to state conclusions about the cognitive and psychological health of the patient group. The online assessment limited the selection of the screening measures, and the normative values for the cognitive tests were obtained in an unconventional person-to-person testing situation. Given the lack of a structured assessment prior to the infection, we could not clarify the impact of SARS-CoV-2 infection on adolescents’ cognitive performances and mental health.

## 5. Conclusions

We found that cognitive, psychological, and somatic symptoms may be very common among the adolescents recovered from COVID-19. Our data reveal a substantial prevalence of depression and anxiety in adolescents following COVID-19 infection. The symptoms, such as fatigue, brain fog, or self-reported memory problems, may be associated with both anxiety and depression, but also with low performance on memory or executive function tasks. Because of the potential impact that SARS-CoV-2 infection may have on adolescents’ long-term health, a systematic evaluation of adolescents recovered from COVID-19 should include both a neuropsychological and a psychological screening.

## Figures and Tables

**Figure 1 brainsci-12-00969-f001:**
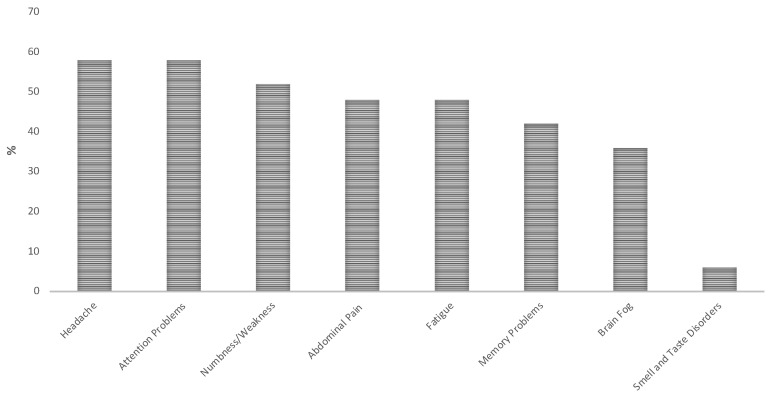
Reported long-COVID Symptoms.

**Table 1 brainsci-12-00969-t001:** Percentage of subjects with neuropsychological difficulties (Panel A) and with depression and anxiety symptoms (Panel B).

**Panel A**					
		**Clinical, *n* (%)**	**Borderline, *n* (%)**		**Normal, *n* (%)**
**SDMT-Attention and processing speed**		1 (3.2)	2 (6.5)		28 (90.3)
**BVN 12–18—Short term memory**					
Forward digit span		0 (0)	11 (35.5)		20 (64.5)
Backward digit		0 (0)	5 (16.1)		26 (83.8)
**BVN 12–18—Long term memory**					
Word immediate recall		4 (12.9)	5 (16.1)		22 (71)
Word delayed recall		4 (12.9)	1 (3.2)		24 (83.9)
**BVN 12–18—Executive functions**					
Category fluency		3 (9.7)	14 (45.2)		14 (45.2)
Phonemic fluency		6 (19.3)	10 (32.3)		15 (48.4)
**Panel B**					
	**No symptoms (%)**	**Positive scores (%)**	**Mild (%)**	**Moderate (%)**	**Severe (%)**
**PHQ-9, *n* (%)**	6 (19.3)	25 (80.6)	15 (48.4)	7 (22.6)	3 (9.7)
**GAD-7, *n* (%)**	12 (38.7)	19 (61.3)	9 (29.0)	9 (29.0)	1 (3.3)

## Data Availability

The datasets generated for this study are available on request to the corresponding author.

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
