# Peer review of "Cognitive Difficulties, Psychological Symptoms, and Long Lasting Somatic Complaints in Adolescents with Previous SARS-CoV-2 Infection: A Telehealth Cross-Sectional Pilot Study"

_brainsci, 2022, doi:10.3390/brainsci12080969_

Round 1

Reviewer 1 Report

The article contains important information. Unfortunately, it is not original and confirms generally known results. It seems to me that by using the same data, the authors could present (or at least give more emphasis to) those issues that are new insights or conclusions. In my opinion, the statistical analysis procedures are incomplete. I am missing information on what statistical analysis tests the authors used? I understand that the authors performed correlation analyses? Only significance levels are given, but not specific information about the procedure performed.

Author Response

The article contains important information. Unfortunately, it is not original and confirms generally known results. It seems to me that by using the same data, the authors could present (or at least give more emphasis to) those issues that are new insights or conclusions. We would like to thank the Reviewer for his/her comments on the manuscript. Data on cognitive functioning in pediatric subjects recovered from SARS-CoV-2 infection are sparse. To highlight the novelty of our research, we modified the sequence of the variables explored (cognitive difficulties, psychological problems and long-COVID complaints) in the whole paper. We change the title as follows “Cognitive difficulties, psychological symptoms, and long lasting somatic complaints in adolescents with previous SARS-CoV-2 infection: a telehealth cross-sectional pilot study”. Moreover, in order to give more emphasis to the new insights of the study, we modified the aims of the study: “Our study aimed to investigate the prevalence of neuropsychological difficulties, psychological symptoms and self-reported long-COVID complaints in a sample of adolescents (3-6 months after recovery from infection) using a screening procedure via telehealth. We studied the possible differences in cognitive and psychological functioning: 1) between adolescents who had symptoms during the infection and those who were asymptomatic; and 2) between adolescents who reported long-COVID symptoms and those who did not. We hypothesized that: 1) adolescents recovered from SARS-CoV-2 infection would report a high prevalence of neuropsychological difficulties, psychological symptoms and long-COVID complaints; 2) cognitive difficulties, psychological symptoms and self-reported long-COVID complaints would be higher in subjects reporting symptoms during the infection; 3) neuropsychological performance would be lower in adolescents who described symptoms during the infection and in those who reported at least one long-COVID symptom; on the other hand, we hypothesized higher psychological symptoms in these adolescents’ subgroups.” In the introduction, we added the sentence “No studies analyzed the possible association between cognitive and psychological problems with self-reported long-COVID complaints among adolescents” (line 64). In the discussion, we wrote the sentence “This is the first study to compare the cognitive performance and psychological symptoms between adolescents with and without long-COVID complaints” (line 299). Moreover, in the discussion we added: “Our data confirmed previous literature showing that long-COVID complaints may be very common in pediatric age [6-8]” (line 264).

In my opinion, the statistical analysis procedures are incomplete. I am missing information on what statistical analysis tests the authors used? I understand that the authors performed correlation analyses? Only significance levels are given, but not specific information about the procedure performed.

To address the concerns raised by the Reviewer, we improved the paragraph on statistical methods. Moreover, in the results the following sentence “Moreover, T test analysis showed statistical differences in cognitive performance and scores on mental health scales between long-COVID and no long-COVID subgroups” was added (line 235).

Reviewer 2 Report

Review of: Long lasting somatic complaints, psychological symptoms, and cognitive difficulties in adolescents with previous SARS-CoV-2 infection: a telehealth pilot study.

Manuscript ID   brainsci-1793855

A small sample of adolescents (N=31), infected with COVID-19 within 3-6 months of the study were examined in this pilot study. The aim of the study was threefold: 1) to investigate the prevalence of self-reported long-COVID symptoms (somatic, mental health, and cognitive); 2) compare symptoms between those who were symptomatic after infection and those who were asymptomatic, and 3) to examine the association between long-COVID complaints, mental health and cognitive problems.

The following are my comments (major and minor comments combined):

Abstract: The abstract is clear and includes all the relevant information. However, there is no mentioning of the major limitation of the study which is the lack of a control group.

Introduction: The introduction is clear and generally it reads well. However, the sentence in line 66-67 disrupts the flow and is out of place and should be removed.

In the introduction it is stated that “the true prevalence of pediatric long-COVID is so far not determined” and it is not clear how this pilot study will change the picture. This study does not clarify outstanding issues regarding pediatric long-COVID.

Method: The method is clear and there is an ethical statement. In line 123 there is a statement on BVN 12-18 raw score conversion. Is this in the correct place? It is unclear to me which of the tests listed are a part of the BVN 12-18. Perhaps it would be more appropriate to list first which subtests of the BVN 12-18 were used and then mention raw score conversion. Regarding the use of PHQ-9 and GAD-7 I find the references inappropriate. The PHQ-9 has been modified for adolescents (PHQ-A) and that is version that should be referred although it may not greatly differ from the original. As it stands, it is unclear which version was used. For the GAD-7 perhaps a paper using the scale in an adolescent sample should also be referenced?

Line 146 – 147: Is something missing? GAD-7 measures anxiety symptoms so the sentence should perhaps read “……with higher scores indicating more severe depression and anxiety symptoms”.

I would like to mention that the testing conditions are not standard and therefore a well matched control group is necessary in order to be able to state anything about the psychological/cognitive health of the patient group. The normative values for the cognitive tests are undoubtedly obtained in a conventional person-to-person testing situation but not across the internet.

Results: Figure 1: there is no title but a text fragment that belongs in the main text (highlighted in pdf).

Table 1: Why is Table 1 presented in two parts, i.e. panel A and B? The % sign is missing in the first line of panel B.

Word usage is somewhat illogical and unclear line 196-197 (neuropsychological performance and psychological scores). Should psychological score be scores on mental health scales? See also line 209: “….in the psychological and neuropsychological profile”. This should be clarified as psychology certainly falls under neuropsychology.

Discussion/Conclusion: 

Line: 220 is a very weak introduction to the discussion as it relates to the current study.

Some discussion about the prevalence of headaches, anxiety and depression in a non-COVID sample is missing. This is critical as there is no control group in the study. The same applies to suicidal ideation which has been, in some studies of adolescents, been found to be quite high. The suicidal ideation in the group studied here is, for example, very close to what was seen in Souza et al (2009) https://www.scielo.br/j/rbp/a/g4cKKLwMvBKtJNV3yJXWN5s/?format=pdf&lang=en. They reported a prevalence of 14.1%.

In addition to considering the general adolescent population in the discussion, which is not done, the importance of considering non-COVID factors, such as general stress related to the pandemic is highly relevant. This is done in the discussion chapter (starting line 270). This is good and further shows the importance of having a control group.

In the final lines of the discussion the authors acknowledge the limitations of their study (e.g., small sample, no control group). Yet, in the conclusion, they immediately go on to making more sweeping claims than they should do given those limitations.

English language and proofreading: generally good, some suggestions in pdf file though.

Author Response

Abstract: The abstract is clear and includes all the relevant information. However, there is no mentioning of the major limitation of the study which is the lack of a control group.

We thank the Reviewer for the suggestion. In the conclusion of the abstract, we added “This is a pilot study limited by the lack of control group. However, we found that cognitive, psychological, and physical symptoms were very common among adolescents recovered from COVID-19”.

Introduction: The introduction is clear and generally it reads well. However, the sentence in line 66-67 disrupts the flow and is out of place and should be removed.

We apologize for our carelessness. The sentence missed the word “telehealth” (“With the spread of the SARS-CoV-2 pandemic telehealth have come innovative solutions in terms of prevention, screening, triage and follow-up”). However, we delated it.

In the introduction it is stated that “the true prevalence of pediatric long-COVID is so far not determined” and it is not clear how this pilot study will change the picture. This study does not clarify outstanding issues regarding pediatric long-COVID. We delated the sentence.

Method: The method is clear and there is an ethical statement. In line 123 there is a statement on BVN 12-18 raw score conversion. Is this in the correct place? It is unclear to me which of the tests listed are a part of the BVN 12-18. Perhaps it would be more appropriate to list first which subtests of the BVN 12-18 were used and then mention raw score conversion.

We agree with the Reviewer opinion. In our revised manuscript, we added “BVN 12-18” also in the paragraph on executive functions (“- Executive functions. The “Verbal Fluency” subtest of BVN 12-18 evaluates executive functions.”) (line 120). Moreover, the statement on raw score conversion was replaced at the end of the paragraph on cognitive measures (line 131). We apologize for having reported “T” scores instead of “standard” scores.

Regarding the use of PHQ-9 and GAD-7 I find the references inappropriate. The PHQ-9 has been modified for adolescents (PHQ-A) and that is version that should be referred although it may not greatly differ from the original. As it stands, it is unclear which version was used. For the GAD-7 perhaps a paper using the scale in an adolescent sample should also be referenced?

We deleted the reference n 36. and we added the following “Kroenke, K.; Spitzer, R.L.; Williams, J.B. The PHQ-9: validity of a brief depression severity measure. J Gen Intern Med 2001, 16(9), 606-613. doi: 10.1046/j.1525-1497.2001.016009606” (Ref. 34). Moreover, we added the reference “Zhou, S.J.; Zhang, L.G.; Wang, L.L.; Guo, Z.C.; Wang, J.Q.; Chen, J.C.; Liu, M.; Chen, X.; Chen, J.X. Prevalence and socio-demographic correlates of psychological health problems in Chinese adolescents during the outbreak of COVID-19. European child & adolescent psychiatry 2020, 29(6), 749-758. https://doi.org/10.1007/s00787-020-01541-4” (Ref. 35).

Line 146 – 147: Is something missing? GAD-7 measures anxiety symptoms so the sentence should perhaps read “……with higher scores indicating more severe depression and anxiety symptoms”. As suggested by the Reviewer, we added “anxiety” in the sentence (line 149).

I would like to mention that the testing conditions are not standard and therefore a well matched control group is necessary in order to be able to state anything about the psychological/cognitive health of the patient group. The normative values for the cognitive tests are undoubtedly obtained in a conventional person-to-person testing situation but not across the internet.

We thank the Reviewer for raising this critical issue. We have modified the limits of the study and have mentioned these possible biases. We wrote: This study had several limitations. It included a small sample size and there was not a matched control group, therefore the study may not be representative of the whole population of adolescents recovered from SARS-CoV-2 infection. In particular, the absence of a control group may represent a selection bias in order to be able to state conclusions about the cognitive and psychological health of the patient group. The online assessment limited the selection of the screening measures and the normative values for the cognitive tests were obtained in a not conventional person-to-person testing situation. Given the lack of a structured assessment prior to the infection, we could not clarify the impact of SARS-CoV-2 infection on adolescents’ cognitive performances and mental health”.

Results: Figure 1: there is no title but a text fragment that belongs in the main text (highlighted in pdf).

We are sorry, we think there was a mistake in the PDF file. The original word file had the following title “Figure 1. Reported long-COVID Symptoms”.

Table 1: Why is Table 1 presented in two parts, i.e. panel A and B? The % sign is missing in the first line of panel B. We modified the table. Word usage is somewhat illogical and unclear line 196-197 (neuropsychological performance and psychological scores). Should psychological score be scores on mental health scales? See also line 209: “….in the psychological and neuropsychological profile”. This should be clarified as psychology certainly falls under neuropsychology.

We agree with the Reviewer, and we have modified the sentence: “Moreover, T test analysis showed statistical differences in cognitive performance and scores on mental health scales between long-COVID and no long-COVID subgroups” (line 235).

Discussion/Conclusion: Line: 220 is a very weak introduction to the discussion as it relates to the current study.

We thank the Reviewer for the comment. We modified the first part of the discussion as follows: “This cross-sectional pilot study investigated the prevalence of neuropsychological difficulties, psychological symptoms and self-reported long-COVID complaints, in adolescents affected by SARS-CoV-2 infection, using a telehealth screening procedure. Overall, this telehealth screening procedure was both feasible and considered helpful by subjects. This confirmed the current telehealth literature providing consistent evidence of benefits for communication and counseling [39, 40]. Increasing studies have demonstrated telehealth’s feasibility and a high level of satisfaction from providers, patients, and families [41]”

Some discussion about the prevalence of headaches, anxiety and depression in a non-COVID sample is missing. This is critical as there is no control group in the study. The same applies to suicidal ideation which has been, in some studies of adolescents, been found to be quite high. The suicidal ideation in the group studied here is, for example, very close to what was seen in Souza et al (2009) https://www.scielo.br/j/rbp/a/g4cKKLwMvBKtJNV3yJXWN5s/?format=pdf&lang=en. They reported a prevalence of 14.1%.

We thank the Reviewer for raising this point. As suggested, we discussed about the prevalence of headaches, anxiety, and depression in a non-COVID sample; moreover, we added a sentence on suicidal ideation in non-COVID and reported the suggested reference. We added: “Moreover, somatic complaints such as headache, together with anxiety and mood disorders are very common also in non-COVID samples. A nationwide Italian study, conducted shortly before the outbreak of the pandemic, reported recurrent headache in almost 45% among the population (15-year-old adolescents) [50]. As for mental health, over the last decades, several studies described an increasing risk of depression and anxiety among children and young people [51, 52]. As reported in 2017 by WHO de-pressive disorders and anxiety are respectively the third and the fifth most frequent cause of adolescent disability-adjusted life-years lost. Literature data on non-COVID samples described a relatively high rate of suicidal ideation. In a study of Souza et al., suicidal ideation in early adolescence showed a rate of 14.1%, which is very close to the rate found in our study [53].

Studies comparing COVID samples to non-COVID ones are required in order to better understand the impact of COVID on cognitive performances and psychological symptoms” In addition to considering the general adolescent population in the discussion, which is not done, the importance of considering non-COVID factors, such as general stress related to the pandemic is highly relevant. This is done in the discussion chapter (starting line 270). This is good and further shows the importance of having a control group. In the final lines of the discussion the authors acknowledge the limitations of their study (e.g., small sample, no control group). Yet, in the conclusion, they immediately go on to making more sweeping claims than they should do given those limitations.

We modified the sentence into the following: “We found that cognitive, psychological, and somatic symptoms may be very common among adolescents recovered from COVID-19” (line 358).

English language and proofreading: generally good, some suggestions in pdf file though. We are sorry, but we could not find the PDF file. However, our paper underwent extensive English editing.

Reviewer 3 Report

Review for Brain Sciences, Manuscript ID: brainsci-1793855: Long lasting somatic complaints, psychological symptoms, and cognitive difficulties in adolescents with previous SARS-CoV-2 infection: a telehealth pilot study

The authors present data concerning somatic complaints, psychological symptoms, and cognitive difficulties in adolescents with previous SARS-CoV-2 infection.  Strengths of the paper include the topic of interest for readers and good presentation and writing.  The main weakness is that it is hard to pinpoint what, if anything, the paper actually contributes toward scientific advancement, because of limited sample size, lack of control group, lack of known prior history of study participants, potential for selection bias, recent increases in depression among adolescents in general, and the well-known association between depression and somatic complaints and anxiety.

However, the authors do a good job of listing limitations, and throughout the paper are cautious not to make unwarranted claims, with the exception of the last sentence of the abstract, which reads “Our study demonstrates that physical, cognitive, and psychological symptoms are very common among adolescents recovered from COVID-19.”  The authors could get the same point across to readers without making unsupported claims.

My only suggestion is that the authors rewrite the last sentence of the abstract to something like “We found that physical, cognitive, and psychological symptoms were very common among adolescents recovered from COVID-19.”  The readers can draw their own conclusions concerning how to interpret this study’s findings.

Author Response

The authors present data concerning somatic complaints, psychological symptoms, and cognitive difficulties in adolescents with previous SARS-CoV-2 infection. Strengths of the paper include the topic of interest for readers and good presentation and writing. The main weakness is that it is hard to pinpoint what, if anything, the paper actually contributes toward scientific advancement, because of limited sample size, lack of control group, lack of known prior history of study participants, potential for selection bias, recent increases in depression among adolescents in general, and the well-known association between depression and somatic complaints and anxiety. However, the authors do a good job of listing limitations, and throughout the paper are cautious not to make unwarranted claims, with the exception of the last sentence of the abstract, which reads “Our study demonstrates that physical, cognitive, and psychological symptoms are very common among adolescents recovered from COVID-19.” The authors could get the same point across to readers without making unsupported claims. My only suggestion is that the authors rewrite the last sentence of the abstract to something like “We found that physical, cognitive, and psychological symptoms were very common among adolescents recovered from COVID-19.” The readers can draw their own conclusions concerning how to interpret this study’s findings.

Thank you for these comments, we agree with the Reviewer and added in the abstract the sentence suggested.

Round 2

Reviewer 1 Report

I thank the authors for addressing the comments and improving the article. I accept the clarifications. I still believe that self-reports on cognitive aspects of activity may differ from those made in experimental studies - and the results of the latter are published. And in this context, the article seems unoriginal to me. However, with the corrections made, I view it positively.

Reviewer 3 Report

The authors have addressed my comments.